# Interlaboratory Validation of a DNA Metabarcoding Assay for Mammalian and Poultry Species to Detect Food Adulteration

**DOI:** 10.3390/foods11081108

**Published:** 2022-04-12

**Authors:** Stefanie Dobrovolny, Steffen Uhlig, Kirstin Frost, Anja Schlierf, Kapil Nichani, Kirsten Simon, Margit Cichna-Markl, Rupert Hochegger

**Affiliations:** 1Austrian Agency for Health and Food Safety (AGES), Department for Molecular Biology and Microbiology, Institute for Food Safety Vienna, Spargelfeldstrasse 191, 1220 Vienna, Austria; stefanie.dobrovolny@ages.at; 2QuoData GmbH, Prellerstrasse 14, 01309 Dresden, Germany; steffen.uhlig@quodata.de (S.U.); kirstin.frost@quodata.de (K.F.); anja.schlierf@quodata.de (A.S.); kapil.nichani@quodata.de (K.N.); kirsten.simon@quodata.de (K.S.); 3Department of Analytical Chemistry, Faculty of Chemistry, University of Vienna, Währinger Strasse 38, 1090 Vienna, Austria

**Keywords:** DNA metabarcoding, animal species, species identification, NGS, food adulteration, validation, interlaboratory ring trial

## Abstract

Meat species authentication in food is most commonly based on the detection of genetic variations. Official food control laboratories frequently apply single and multiplex real-time polymerase chain reaction (PCR) assays and/or DNA arrays. However, in the near future, DNA metabarcoding, the generation of PCR products for DNA barcodes, followed by massively parallel sequencing by next generation sequencing (NGS) technologies, could be an attractive alternative. DNA metabarcoding is superior to well-established methodologies since it allows simultaneous identification of a wide variety of species not only in individual foodstuffs but even in complex mixtures. We have recently published a DNA metabarcoding assay for the identification and differentiation of 15 mammalian species and six poultry species. With the aim to harmonize analytical methods for food authentication across EU Member States, the DNA metabarcoding assay has been tested in an interlaboratory ring trial including 15 laboratories. Each laboratory analyzed 16 anonymously labelled samples (eight samples, two subsamples each), comprising six DNA extract mixtures, one DNA extract from a model sausage, and one DNA extract from maize (negative control). Evaluation of data on repeatability, reproducibility, robustness, and measurement uncertainty indicated that the DNA metabarcoding method is applicable for meat species authentication in routine analysis.

## 1. Introduction

Food authentication is known to be a challenging task. The methodology applied depends on several factors, including sample type, type of adulteration, and the information required. Meat products are most commonly adulterated by the replacement of high-priced animal species by lower-quality or cheaper ones. Since DNA-based methodologies are highly suitable to detect genetic variations such as single nucleotide polymorphisms (SNPs), insertions, and deletions, they play a crucial role in the identification and differentiation of animal species in meat products [1,2,3]. DNA-based methodologies target either species-specific sequences in nuclear DNA or conserved regions in mitochondrial DNA [4]. Currently, authentication of meat products in official food laboratories is mainly based on real-time polymerase chain reaction (PCR) assays and/or DNA arrays. Multiplex real-time PCR assays, allowing the identification and quantification of multiple species in one and the same well, are particularly applicable for routine analysis because they allow saving time and costs. Multiplex real-time PCR assays have not only been developed for domesticated species, e.g., beef, pig, chicken, and turkey [5], and beef, pig, horse, and sheep [6], but also for game species, e.g., roe deer, red deer, fallow deer, and sika deer [7]. However, the low number of optical channels of real-time PCR instruments limits the number of species that can be targeted simultaneously.

Next generation sequencing (NGS) technologies, in particular massively parallel sequencing of PCR products based on the analysis of species-specific differences in DNA sequences (DNA barcoding), is being considered a promising alternative [8,9,10]. So-called DNA metabarcoding offers the possibility to identify a wide variety of species not only in individual foodstuffs but even in complex mixtures. Moreover, in contrast to real-time PCR assays, it is an untargeted approach, allowing the detection of species one has not been looking for.

We have recently developed a DNA metabarcoding method for 15 mammalian species and six poultry species, which are quite frequently contained in European foodstuff [11]. In order to detect both mammalian and poultry species, a primer pair for mammals and a primer pair for poultry species was combined in a duplex PCR assay. A ~120 bp fragment of the mitochondrial 16S ribosomal DNA gene serves as barcode region. The DNA metabarcoding method has been validated with regard to specificity, repeatability, robustness, accuracy, and limit of detection (LOD) [11]. In-house validation data showed that the DNA metabarcoding method can be used for routine applications. Meat species can be identified down to a concentration of 0.1%. Very recently, the applicability of the DNA metabarcoding method for routine analysis was further investigated by the analysis of a total of 104 samples (25 reference samples, 56 food products, and 23 pet food products). Results obtained by DNA metabarcoding were in line with those obtained by real-time PCR and/or a commercial DNA array [12]. However, interlaboratory evaluation of novel methods is a prerequisite for standardization and harmonization.

In this study, we summarized the results of an interlaboratory ring trial for the DNA metabarcoding method, initiated by the §64 German Food and Feed Code (LFGB) working group “NGS Species Identification”, chaired by the Federal Office of Consumer Protection and Food Safety (BVL) in Germany. One goal of the working group is the validation and standardization of (screening) methods for the identification and differentiation of animal species based on next generation amplicon sequencing for food authentication. The interlaboratory ring trial was coordinated by the Austrian Agency for Health and Food Safety (AGES) in 2020 and involved 15 participating laboratories. The aim was to evaluate the performance (e.g., repeatability, reproducibility, accuracy) of the DNA metabarcoding method in detail.

## 2. Materials and Methods

### 2.1. Participating Laboratories

The interlaboratory ring trial was organized by the AGES on behalf of the BVL. The following laboratories participated in the ring trial (in alphabetical order): Bavarian State Office for Food Safety and Health (LGL), Oberschleißheim, Germany; Chemical and Veterinary Analytical Institute Muensterland-Emscher-Lippe (CVUA-MEL), Muenster, Germany; Chemical and Veterinary Investigation Office Freiburg (CVUA-FR), Freiburg, Germany; Chemical and Veterinary Investigation Office Karlsruhe (CVUA-KA), Karlsruhe, Germany; Eurofins Genomics Europe Applied Genomics GmbH, Ebersberg, Germany; StarSEQ GmbH, Mainz, Germany; Labor Kneissler GmbH and Co. KG, Burglengenfeld, Germany; Saxony-Anhalt State Office for Consumer Protection (LAV S-A), Halle, Germany; State Office Laboratory Hessen (LHL), Kassel, Germany; Max Rubner-Institut (MRI)/National Reference Centre for Authentic Food (NRZ-Authent), Kulmbach, Germany; Max Planck Institute for Plant Breeding Research (MPIPZ), Köln, Germany; Lower Saxony State Office for Consumer Protection and Food Safety (LAVES), Niedersachsen, Germany; AGES, Vienna, Austria; PLANTON Laboratory for Analysis and Biotechnology GmbH, Kiel, Germany; SGS Institute Fresenius GmbH, Freiburg, Germany.

### 2.2. Samples

In the course of the interlaboratory ring trial, eight samples had to be analyzed: six DNA extract mixtures (samples 1–6), one DNA extract from a model sausage (sample 7), and one DNA extract from maize (sample 8), serving as a negative control (Table 1).

In total, seven animal species, including five mammalian species (*Sus scrofa domesticus* (pig), *Bos taurus* (cattle), *Equus caballus* (horse), *Ovis gmelini aries* (sheep), and *Capra aegagrus hircus* (goat)), and two poultry species (*Gallus gallus domesticus* (chicken) and *Meleagris gallopavo* (turkey)) were covered by the samples. All samples originated from muscle meat and were purchased from local meat suppliers.

Sample 1 contained DNA from seven animal species: DNA from pig as major component (94%, *v*/*v*), and DNA from six animal species (cattle, horse, sheep, goat, chicken, turkey; 1% (*v*/*v*) each). Samples 2–6 consisted of DNA from five animal species in varying proportions, ranging from 0.1% (*v*/*v*) to 67.5% (*v*/*v*). All DNA extract mixtures were prepared at the AGES.

The model sausage was produced according to the Codex Alimentarius Austriacus by the Higher Technical College for Food Technology Hollabrunn (Hollabrunn, Austria). The model sausage consisted of 50% (*w*/*w*) beef, 40% (*w*/*w*) pig, 5% (*w*/*w*) chicken, and 5% (*w*/*w*) turkey.

### 2.3. Genomic DNA Extraction

Extraction of genomic DNA from muscle meat of the seven animal species (pig, beef, horse, sheep, goat, chicken, turkey) was carried out at the CVUA-FR by applying the official method L 00.00–119 [13]. Identity of the animal species was verified by subjecting DNA extracts to Sanger sequencing and matching a ~464 base pair (bp) fragment of the mitochondrial cytochrome b gene against public databases provided by the National Center for Biotechnology Information (NCBI, Bethesda, MD, USA) [14,15]. After verification by Sanger sequencing and isolating genomic DNA fourfold, individual DNA extracts were combined. Total DNA of the (combined) DNA extracts was quantified by spectroscopy employing a UV/VIS spectrophotometer, adjusted to a DNA concentration of 20 ng/µL and sent to AGES. Isolation of DNA from the homogenized model sausage was performed at AGES [13].

The copy number of the mitochondrial 16S ribosomal DNA gene in the extracts from the respective animal species was determined by droplet digital PCR (ddPCR, QX200 Droplet Generator, QX200 Droplet Reader (Bio-Rad, Hercules, CA, USA)) using the EvaGreen Supermix. DNA extract mixtures were prepared at the AGES, by taking the copy numbers (pig (870 copies/µL), beef (1069 copies/µL), horse (1795 copies/µL), sheep (520 copies/µL), goat (790 copies/µL), chicken (620 copies/µL), turkey (673 copies/µL)), into account. The percentages of samples 1 to 6 given in Table 1 were calculated by relating the DNA copy number of the respective animal species to the total number of copies of animal species in the sample.

### 2.4. Study Design

The interlaboratory ring trial for validation of the DNA metabarcoding method for mammalian and poultry species [11] was conducted in the framework of the §64 LFGB working group “NGS Species Identification” under the coordination of AGES. Statistical data analysis was performed by QuoData GmbH (Dresden, Germany).

For sequencing, three benchtop NGS instruments from two companies were used. Benchtop instruments from Illumina (Illumina, San Diego, CA, USA) were employed by eleven laboratories, whereof eight used the MiSeq instrument, three the iSeq 100 instrument, and one participant used both the MiSeq and the iSeq 100 instrument. Four laboratories applied the Ion GeneStudio S5 instrument from Thermo Fisher Scientific (Thermo Fisher Scientific, Waltham, MA, USA).

Each participant obtained 16 anonymously labelled samples, comprising two subsamples of each of the eight samples (Table 1). Sixteen samples were chosen to allow the iSeq 100 platform to be included in the ring trial. This also enabled the use of the most cost-effective MiSeq Reagent Micro Kit v2 for a small number of samples on the Illumina platforms. Participants directly used all individual DNA extracts for DNA library preparation and subsequently for amplicon sequencing on a next-generation sequencing instrument. Together with the “ready-to-use” DNA extracts, the participants obtained reagents for creating DNA libraries, a sequencing kit, and a step-by-step instruction.

In order to be able to perform sequencing on the Ion GeneStudio S5 instrument (Thermo Fisher Scientific, Waltham, MA, USA), the protocol for preparation of DNA libraries and sequencing published previously by Dobrovolny et al. (2019) [11] had to be adapted as follows. Each of the two forward primers were elongated by a 3 bp barcode adapter, one of 16 different 10 bp barcodes (index sequence) and the overhang adapter sequence (A adapter). Each of the two reverse primers was linked to an overhang adapter sequence (trP1 adapter). The PCR setup did not include additional magnesium chloride solution. In the magnetic bead cleaning step, a total of 37.5 µL Agencourt^®^ AMPure^®^ XP beads (Beckman Coulter, Brea, CA, USA) was used and the DNA was eluted with 50 µL Tris-EDTA (TE) buffer. The average library size was 190 bp, and all DNA libraries were adjusted to 100 pM and were mixed together in a single 1.5 mL tube. A 25 pM DNA pool was used for sequencing. In general, any deviations from the protocols had to be reported by the participants.

Paired-end sequencing on an Illumina instrument was performed using either the MiSeq Reagent Micro Kit v2 (300-cycles) or the iSeq 100 i1 Reagent v2 (300-cycles), which included a 5% PhiX spike-in. The Ion Chef instrument was used with the Ion 510^TM^ and Ion 520^TM^ and Ion 530^TM^ Kit-Chef and the Ion 520^TM^ Chip Kit to perform template preparation, enrichment and chip loading. Finally, the sequencing reaction was started on the Ion GeneStudio S5 instrument.

To obtain information about the presence of the animal species in the samples, the sequencing output in FastQ format was processed with a multi-step analysis pipeline by using Galaxy (version 19.01) as described previously [12]. Before the resulting FastQ files were used as input for the data analysis, the raw binary base call (bcl) files generated by Illumina devices were converted to text files using the conversion software bcl2fastq2-v2.19.0.316 (Illumina, San Diego, CA, USA). The default demultiplexing option of one allowed mismatch in the barcode recognition of the Illumina software (-- barcode mismatches) was thereby set to zero (value: 0) and the step was also integrated into the pipeline. Preliminary tests had shown that this can increase the quality of index recognition. The Thermo Fisher instrument software uses this setting by default. The analysis pipeline for sequencing data of the Thermo Fisher Scientific (Waltham, MA, USA) platforms was modified because paired-end FastQ files do not exist in this case. Consequently, the primer sequences were adapted according to the requirements of the analysis tool Cutadapt (Galaxy version 1.16.6 [16]) and the tool fastq-join (Galaxy version 1.1.2-806.1) [17] was removed. Dereplicated reads were directly matched against a customized database (AGES database) including 51 mitochondrial genomes from animals (Appendix A) and the public databases provided by NCBI using BLASTn [18]. The AGES database contains verified sequences from the NCBI database exclusively from food-relevant animal species. This reduces the time needed for alignment and is intended to avoid nonsense results. For each of the samples, results were listed automatically in a table according to taxonomy and abundance and a formula calculated the proportions of animal species by relating the number of reads for the respective species to the number of total reads (after pipeline) across all animal species obtained for the subsample. For further statistical analysis, all Excel spreadsheets were sent to QuoData GmbH.

### 2.5. Statistical Data Analysis

Statistical analyses were performed by QuoData GmbH. Even though the DNA metabarcoding method used in this interlaboratory comparison is commonly regarded as a qualitative method, the underlying decision process is based on the comparison of a quantitative value, namely the proportion of a single species, with a specific threshold. The performance of such a method can be assessed both on the basis of the qualitative result (yes/no) and on the basis of the underlying quantitative data. Because the information content of the quantitative data can be far greater than the corresponding qualitative data, the quantitative data were used in addition to the qualitative data to describe the performance of the DNA metabarcoding method.

In addition, the study of quantitative data also aimed to verify the extent to which this method can also be used for quantitative determinations.

#### 2.5.1. Quantitative Statistical Analyses

Proportions of animal species ranged between 0.1% and 94%. To avoid asymmetric distributions for proportions near 0% and 100%, and to ensure equality of variances for the individual combinations of samples/animal species, the proportions were subjected to a logit transformation:$$\mathrm{logit}(\mathrm{proportion})=\ln\;\left(\frac{\mathrm{proportion}}{1-\mathrm{proportion}}\right)$$

The logit-transformed proportions can be retransformed as follows:$$\mathrm{proportion}=\frac{e^{\mathrm{logit}(\mathrm{proportion})}}{{1+e}^{\mathrm{logit}(\mathrm{proportion})}}$$

The logit-transformed proportions were then subjected to several outlier tests. Data were checked for systematic errors across samples and/or animal species affecting the mean values (Mandel h statistics) and/or variances (Mandels k statistics). In addition, the occurrence of sample- and animal species-specific outliers regarding the laboratory mean values and variances was tested for by means of the Grubbs and the Cochran tests (significance level 1%), respectively. Proportions identified as outliers were excluded from further statistical analyses.

Logit-transformed and outlier-cleaned data were checked for normal distribution using the Shapiro–Wilk test. Then, repeatability, reproducibility, and accuracy of the proportions of animal species were determined according to the criteria of QuoData certified with the aid of the software solution for method comparison studies and interlaboratory comparison studies PROLab Plus, version 2021.7.22.0 [19] (QuoData, Dresden, Germany), using the statistical methods according to DIN ISO 5725-2 and according to the specifications in the Official Collection of Test Methods ASU §64 LFGB for the statistical evaluation of ring trials for method validation [20]. Taking into account the obtained repeatability and reproducibility standard deviations for samples 1–7, variance functions describing the functional relationship between standard deviations and overall mean for the individual combinations of samples/animal species were modelled.

For each of the combinations, the bias (difference) between this overall mean and the proportion of the animal species added to the sample was also determined. Furthermore, the standard deviation of this bias was calculated.

Prediction profiles and measurement uncertainty profiles were constructed, both not considering the bias (based on reproducibility standard deviations), and considering the bias (based on reproducibility standard deviations as well as on the standard deviation of the bias).

In addition, the z scores for each combination of lab/sample/animal species were determined, providing a measure for the standardized deviations of laboratory mean values from the respective overall mean value.

#### 2.5.2. Qualitative Statistical Analyses

A sample was classified as false positive if for at least one animal species that had not been added, a proportion above a defined threshold was obtained. By contrast, a sample was considered false negative for a specific animal species if the proportion was below a defined threshold for this species.

The probability of detection for an arbitrary animal at a defined threshold for (1) a laboratory with average performance, (2) a laboratory with positive bias, and (3) a laboratory with negative bias was determined based on the variance functions by the quantitative statistical analysis.

## 3. Results and Discussion

Fourteen of the fifteen laboratories submitted their sequencing results in time. Ten laboratories applied the Illumina platform, with seven laboratories (01, 02, 03, 04, 06, 08, 14) using the MiSeq, two laboratories (07, 13) the iSeq 100, and one laboratory utilizing both the MiSeq and the iSeq 100 (referred to as “laboratory 15” and “laboratory 20”, respectively). The remaining four laboratories (09, 10, 11, 12) applied the Ion GeneStudio S5 system from Thermo Fisher Scientific.

Each of the laboratories submitted 16 sequencing results in total (eight samples, two subsamples each), with the exception of laboratories 07, 12, and 13. Laboratory 07 did not provide the result for subsample 8B (negative control), whereas datasets submitted by laboratories 12 and 13 were lacking results for both subsamples of sample 8. With the exception of laboratory 03, sequencing was done by using the test kit provided by the AGES. The suitability of the MiSeq Reagent Kit v2 applied by laboratory 03 had been demonstrated in preliminary experiments.

FastQ data provided by the participating laboratories was evaluated by the AGES by using the analysis pipeline in Galaxy. For identification of animal species, the DNA sequences (reads) were aligned, once with the customized AGES database and once with the NCBI database. Appendix A summarizes the total number of reads for each laboratory, taking into account the results obtained for each of the fourteen subsamples containing animal species (samples 1–7).

Total numbers of raw reads that passed the analysis pipeline were quite different between laboratories. Very low total numbers of raw reads can, for example, be caused by errors during wet-lab activities, e.g., pipetting errors or error rate of DNA polymerase. In addition, problems with adapter- and index-recognition are known to have an impact. Loss of reads or their elimination by pipeline tools can also be caused by errors in PCR amplification of the library (e.g., index hopping), sequencing errors (e.g., inserts, substitutions, or deletions) or insufficient cluster resolution [21]. All these errors may affect the quality of raw data (FastQ file) and thus the number of DNA sequences (reads) after analysis pipeline.

Total numbers of reads obtained with the Ion GeneStudio S5 were significantly higher than those obtained with the MiSeq (*p* < 0.001) and the iSeq 100 (*p* < 0.001). Differences observed between the Illumina and the Thermo Fisher technology are caused by differences in data filtering. The instrument-specific software of the Ion GeneStudio S5 removes datasets of lower quality by filtering before starting the analysis pipeline. Thus, considerably more sequences remain after primary data analysis compared to the instruments from Illumina.

Differences in recoveries, by relating the total number of reads to the number of raw reads before analysis pipeline, between laboratories using the same instrument type (MiSeq, iSeq 100, or Ion GeneStudio S5, Appendix A) hint at differences in the quality of the sequencing run and unintended loss of reads. Significant differences (*p* < 0.001) in recoveries between laboratories using Illumina instruments and those applying the Ion GeneStudio S5 were, however, expected. These differences are caused by the fact that the pipeline of the Ion GeneStudio S5 neither included paired-end sequencing nor a “joining step” as was the case with the Illumina platforms.

### 3.1. Quantitative Evaluation of Ring Trial Data

The aim of quantitative evaluation of ring trial data was to determine average proportions of the animal species that had been added to samples 1–7, and to identify resulting error components within and in between laboratories.

#### 3.1.1. Proportions of Animal Species in Samples 1–7

Proportions of animal species were calculated by relating the number of reads for the respective species to the number of total reads (after pipeline) across all animal species obtained for the sample. Table 2 gives the proportions of animal species determined for sample 1 containing seven animal species (Table 2a), sample 2 (as a representative of samples consisting of five animal species; Table 2b) and sample 7, a model food sample (Table 2c). Results for samples 3, 4, 5, and 6 are shown as stacked bar plots (Figure 1).

Preliminary evaluation of the results indicated that the proportions of animal species determined considerably depended on the sequencing platform/technology applied. Due to the low number of laboratories using the sequencing technology from Thermo Fisher Scientific, only results obtained by laboratories applying Illumina platforms were included into statistical evaluation. Results obtained by laboratories 09–12 using the Ion GeneStudio S5 are only shown for comparison.

#### 3.1.2. Logit Transformation

Proportions of animal species in samples 1–7 were quite different, ranging from 0.1% to 94%. However, a prerequisite for the evaluation of ring trial data according to ASU §64 LFGB is that the proportions of animal species follow normal or at least symmetric distribution. In order to allow assumption of normal distribution and ensure equality of variances of the individual combinations of samples/animal species (after elimination of outliers), proportions of animal species were subjected to a logit transformation. The logit is the logarithm of the proportion of the animal species divided by 1 minus the proportion of the animal species. Proportions of, e.g., 0.1%, 1%, 10%, 30%, 50%, and 70%, resulted in logit values of −6.91, −4.60, −2.20, −0.85, 0, and 0.85, respectively. Since the logit for 0% and 100% is not defined, it was set to surrogate values of −15 and +15, respectively.

#### 3.1.3. Outlier Tests

In the course of evaluating ring trial data according to ASU §64 LFGB, logit-transformed proportions of animal species were subjected to several outlier tests (see also Section 2.5.1). Table 3 summarizes the outliers and reasons for their elimination.

In case an outlier was only identified for one database (either AGES or NCBI database), it was, however, eliminated for both databases to ensure data comparability. In total, 15 of 396 (3.8%) combinations of laboratory/sample/animal species were identified as outliers and excluded from further evaluation for each of the databases.

#### 3.1.4. Distribution of Sample-Specific Proportions of Animal Species

After outlier elimination, sample-specific logit-transformed proportions of animal species were tested for normal distribution by Kernel density estimation and the Shapiro–Wilk test.

A small number of cases were found to have a bimodal distribution. However, the Shapiro–Wilk test did not show evidence for non-normality for any of the combinations of samples/animal species. Thus, logit-transformed proportions of animal species could be subjected to further statistical evaluation.

#### 3.1.5. Statistical Parameters According to ASU §64 LFGB

Logit-transformed and outlier-cleaned data were normally distributed and thus could be subjected to statistical evaluation according to ASU §64 LFGB. Table 4 gives—for each animal species and both databases (AGES and NCBI)—main statistical parameters, including re-transformed mean value, reproducibility standard deviation (s_R_), and repeatability standard deviation (s_r_). For the sake of completeness, the logit-transformed parameters are shown as well. The reproducibility standard deviation characterizes the variability of results between laboratories, and the repeatability standard deviation the variability within a laboratory under constant conditions, i.e., the variability of results obtained for the two subsamples of the same sample.

#### 3.1.6. Dependence of Bias, Reproducibility Standard Deviation, and Repeatability Standard Deviation on the Mean Proportion of Animal Species

Next, it was evaluated whether the bias between the proportion of the animal species added to the sample and the overall mean determined in the ring trial, as well as whether reproducibility standard deviation and repeatability standard deviation depended on the proportion of the respective animal species (Figure 2A) and/or the predominant animal species in the sample (Figure 2B). Evaluation was based on the statistical parameters for logit-transformed proportions of animal species.

As expected, the reproducibility standard deviation and repeatability standard deviation were found to be higher for lower proportions of animal species than for proportions of about 50% (logit = 0), independent of the database selected for alignment. It was found that pig tends to have higher standard deviations in reproducibility, but not in repeatability, compared to other animal species (Figure 2A). A tendency towards higher reproducibility standard deviations was also observed in case beef was the predominant animal species in the sample. This also held true for the repeatability standard deviation, although to a lower extent.

The bias between the proportion of the animal species added to the sample and the overall mean determined in the ring trial was in the range from −0.6 to 0.6 logits, with just two exceptions. Neither the animal species nor the proportion of the animal species was found to have a systematic effect on the bias. The animal species and the proportion of the animal species did not have a systematic impact on the reproducibility standard deviation either.

Thus, the standard deviation induced by the bias (“bias standard deviation”), absolute reproducibility standard deviation, and absolute repeatability standard deviation could be modeled across animal species and samples, for both the AGES and NCBI databases. The modeled variance function was similar for both databases. The lowest bias standard deviation, reproducibility standard deviation, and repeatability standard deviation were found for a proportion of 50% (logit = 0). The closer the proportion to 0% or 100%, the higher the standard deviations. Table 5 summarizes the modeled and re-transformed standard deviations, which were found to be independent of the database used for alignment.

#### 3.1.7. Variability across Animal Species and Measuring Uncertainty

To allow predictions for further analyses, the measuring uncertainty was evaluated. Since the database (AGES or NCBI) was not found to have an impact on bias standard deviation, reproducibility standard deviation, or repeatability standard deviation, the measuring uncertainty was only evaluated for the AGES database, representative for both databases.

Based on the reproducibility standard deviation, a prediction profile was established in terms of the 95% confidence interval of the results of all laboratories across all animal species. In addition, the 95% confidence interval was established by considering both the reproducibility standard deviation and the bias standard deviation.

The upper part of Figure 3 shows the 95% confidence interval of the (outlier-cleaned) results depending on the respective overall mean value of the proportion of an animal species (without considering the bias). The left side of the figure shows the entire range, and the right side an enlarged view of proportions from 0 to 10%. The figure indicates that for a “true proportion” (assuming that the overall mean across laboratories applying Illumina platforms reflects the “true proportion”) of, e.g., 5%, the 95% confidence interval is 4.1–6.2%. In total, 4.7% of the individual values are outside the 95% confidence interval.

The 95% confidence interval of the (outlier-cleaned) results of all laboratories, depending on the respective proportion added (by considering the bias) is shown in the lower part of Figure 3. The left and right sides show the entire range and an enlarged view of proportions from 0 to 10%, respectively. For example, for an added proportion of 5%, the 95% confidence interval is 2.5–9.9%. In total, 3.7% of the individual values are outside the 95% confidence interval.

From the prediction profiles, a measurement uncertainty profile was established, indicating how far the proportion determined may deviate from the “true“ proportion of the animal species. Figure 4 shows the 95% measurement uncertainty intervals depending on the proportion of the animal species determined, on the left side without consideration of the bias (assuming that the “true” proportion equals the overall mean across laboratories applying Illumina platforms), on the right side under consideration of the bias (assuming that the “true” proportion equals the proportion of animal species added to the sample).

For a 50% proportion of the animal species, the “true value” can be 3.6% (percentage points) lower or higher, if the bias is not taken into account, or even 15.2%, if the bias is taken into consideration.

#### 3.1.8. z Scores

z scores were calculated for interlaboratory evaluation across samples and animal species (Figure 5). z scores measure standardized deviations of laboratory mean values from the overall mean value. An absolute z score > 2 hints at a statistically significant deviation of the respective laboratory.

LASSO regression was applied to check whether absolute z scores depended on the instrument (MiSeq, iSeq 100), NGS experience of the respective laboratory, and/or activation of the function ”adapter trimming“ before starting the sequencing run. Analyses were performed excluding data previously identified as outliers. NGS experience of the laboratory was found to significantly affect the z score. For laboratories more experienced in NGS (laboratories 01, 04, 08, 15, 20), lower absolute z scores were determined compared to those with lower NGS experience. By contrast, neither the instrument (MiSeq, iSeq 100) nor activation of the function “adapter trimming” before starting the sequencing run was found to significantly affect the z score.

### 3.2. Qualitative Evaluation of Ring Trial Data

#### 3.2.1. False Positive Rate

Next, it was investigated whether animal species were identified that had not been added to the samples, and whether the false positive rate depended on the database selected for alignment. A sample was classified as positive if for at least one animal species that had not been added, a proportion above a defined threshold was obtained. The threshold was set to 0.1%, 0.5%, 1.0%, or 2.0% proportion of the animal species.

Table 6 indicates that alignment against the AGES database (Table 6A) resulted in less false positive reads compared to alignment against the NCBI database (Table 6B).

Alignment against the AGES database only yielded false positive samples at threshold values of 0.05% or 0.1% (Figure 6).

Higher false positive rates for the NCBI database were inevitably caused by the higher number of entries in the NCBI database compared to the AGES database. Most species resulting in false positive reads when using the NCBI database were not contained in the AGES database and thus could not be identified with the latter.

From the ring trial data it can be concluded that by using an appropriate customized database and by setting the threshold to 0.5%, false positive rates < 1% will be obtained.

#### 3.2.2. False Negative Rate

Next, the false negative rate was evaluated at threshold values of 0.05%, 0.1%, 0.5%, and 1% for both the AGES and the NCBI databases (Table 7).

There was no considerable difference between the AGES and the NCBI databases regarding the proportion of false negative results obtained for samples 1–7. At a threshold of 0.05%, false negative results were only obtained for pig in sample 4. At a threshold of 0.1%, none of the combinations of sample–animal species led to false negative results, with the exception of proportions being close to the threshold (0.1%). The data indicate that at a threshold of ≥0.5%, the probability of obtaining false negative results is very low.

#### 3.2.3. Probability of Detection (Qualitative Evaluation)

Figure 7 shows the probability of detection for three thresholds (0.1%, 0.5%, and 1%) and three scenarios, namely a laboratory with average performance, a laboratory with positive bias, and a laboratory with negative bias.

Figure 7 indicates that a threshold value of 1% seems to be a good compromise, provided that the variance function determined in the ring trial is equal to the actual variance function. A threshold value of 1% guarantees that a laboratory with a positive bias (overestimating the actual proportion) does not complain if the proportion of a certain animal species is 0.5%, whereas even a laboratory with a negative bias (underestimating the actual proportion) will be able to identify proportions >1.5% reliably.

### 3.3. Negative Control

Sample 8 was a DNA extract from maize, serving as a negative control. Since the marker system designed for mammals and poultry species does not detect maize, all reads that were obtained for sample 8 had to be regarded as false positive.

Table 8 lists the number of total reads (after pipeline) per laboratory and species. Laboratories 12 and 13 did not submit results for sample 8, laboratory 07 only provided results for one of the two subsamples.

In total, eight animal species were identified in the negative control by alignment against the AGES database. Fourteen further animal species, including the species *Homo sapiens* were identified, when the NCBI database was used. Per laboratory, up to five animal species were only identified with the NCBI database, with the exception of laboratory 06, which even detected eight additional animal species in subsample A.

In most cases, within a laboratory, the number of reads per animal species was similar for both subsamples. When the AGES database was used for alignment, most reads were assigned to beef and pig. Sample multiplexing in general, together with an inappropriate index layout, carries the risk of index misassignment. This is obviously the reason for the over-represented number of reads for pig and beef in the negative controls, as these animal species represent the main quantities in the samples. Although the index kit was used according to the manufacturer’s instructions, the number of reads of these animal species could be reduced to the expected level in a supplementary experiment with an alternative index layout. Alignment against the NCBI database also resulted in considerably high numbers of reads (laboratories 06, 07, and 14 > 300, laboratory 07 even > 1000) for *Homo sapiens*, which was not contained in the AGES database.

As mentioned above, the marker system applied does not detect maize. Thus, the high number of reads for maize obtained by laboratory 06 for one subsample seems to be caused by a random error.

In general, the total number of false reads obtained for both subsamples was similar within a laboratory. Larger differences between the total reads of subsamples was observed for laboratories 03 and 14 (AGES and NCBI databases) and laboratory 06 (only NCBI database).

## 4. Conclusions

In summary, evaluation of data from the interlaboratory ring trial indicates that the DNA metabarcoding method performed on an Illumina platform is applicable for determining the proportion of the seven animal species with the given precision. Furthermore, the applicability of the method for testing foodstuff was demonstrated by the correct identification of the ingredients of a model sausage, which also supports the results in our study published recently.

Based on the data of the ring trial, a threshold of 0.5% is suitable to reliably assess whether a certain animal species is contained in a sample. The DNA metabarcoding method turned out to be rather robust and is therefore suitable to be implemented in routine analysis in official food control laboratories. Even laboratories that did not have much experience in NGS were able to provide reliable results. We suggest strictly following the given protocol. The results of the interlaboratory ring trial indicate that even alternative test kits or various sequencing platforms might be applied. However, the impact of any deviations from the experimental conditions has obviously to be tested before implementation in routine analysis.

Correct index recognition is of particular importance for pooled DNA libraries. We recommend frequently changing the index kits or the use of longer index sequences to avoid false positive and/or false negative results.

For taxonomic assignment, we suggest applying a customized database, as the pipeline is completed significantly faster and no nonsense results from erroneous database entries occur. However, if unexpected read losses and non-identifiable reads occur, the additional use of the entire NCBI database or any other appropriate sequence database is recommended.

In order to increase interlaboratory comparability of results obtained by DNA metabarcoding methods, it would be necessary to establish a reference database with verified sequence entries of relevant species. Access to adequate reference material would also facilitate harmonization of the methods used.

In general, determination of the meat content (*w*/*w*) from the number of NGS reads or the determined target DNA concentration is a well-known difficulty, especially in the quantification of meat species in processed foods. The result is also influenced by the degree of processing of the sample present and by the type of animal ingredients used. Data from testing reference samples out of proficiency testing schemes confirm the limitations known for DNA quantification in meat products [12]. Quantitative results should therefore serve only as rough estimates for weight ratios of different species in food.

## Figures and Tables

**Figure 1 foods-11-01108-f001:**
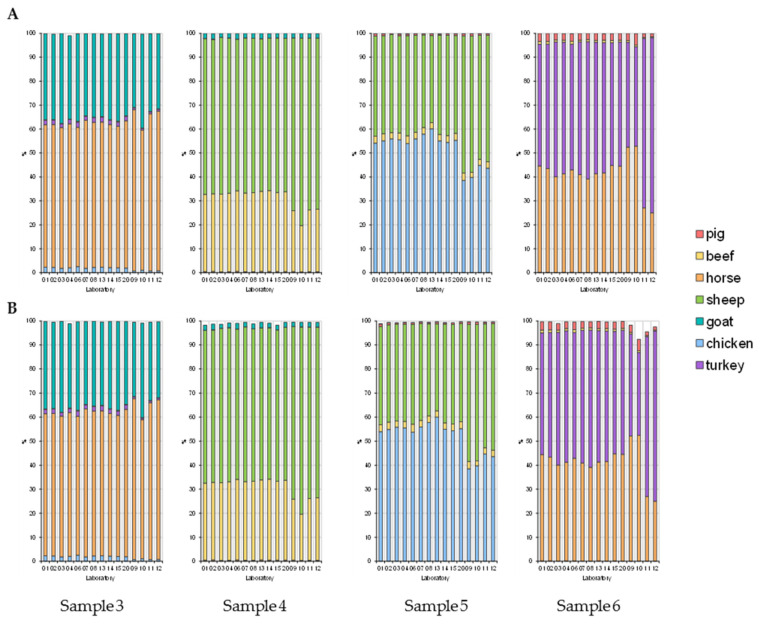
Proportions of animal species determined for samples 3, 4, 5, and 6. (**A**) AGES database, (**B**) NCBI database. Laboratories 01–06, 08, 14, 15: MiSeq; laboratories 07, 13, 20: iSeq 100; laboratories 09–12: Ion GeneStudio S5.

**Figure 2 foods-11-01108-f002:**
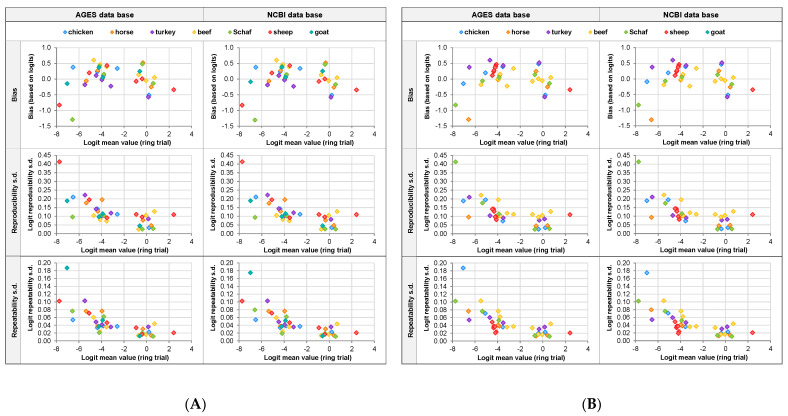
Bias, reproducibility standard deviation, and repeatability standard deviation depending on the overall mean determined in the ring trial based on logit-transformed proportions of animal species. Colors refer to (**A**) the respective animal species and (**B**) the predominant animal species in the respective sample.

**Figure 3 foods-11-01108-f003:**
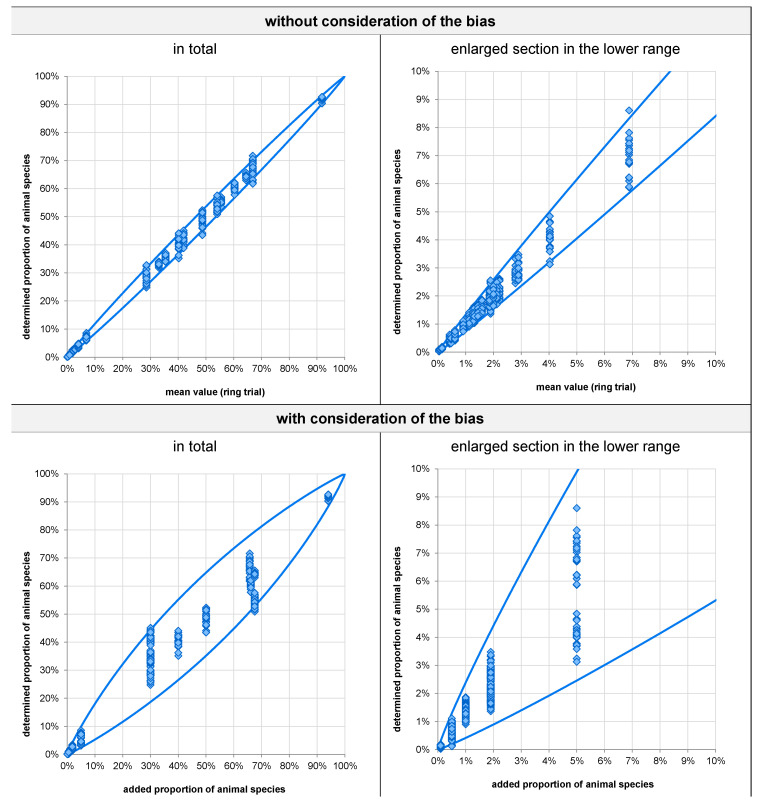
95% confidence interval for the mean (**top**) and added (**bottom**) proportion of animal species based on a single measurement, independent of the animal species.

**Figure 4 foods-11-01108-f004:**
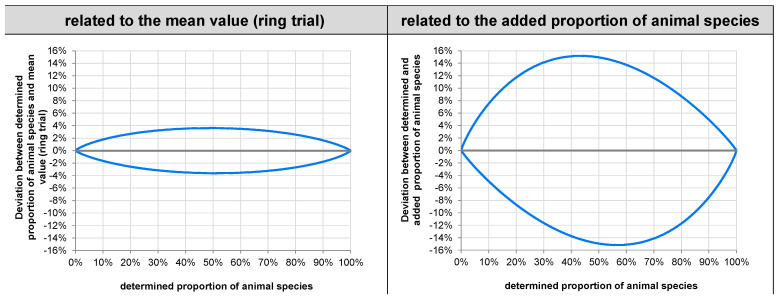
95% measuring uncertainty interval for the proportion of the animal species determined, based on a single measurement, independent of the animal species.

**Figure 5 foods-11-01108-f005:**
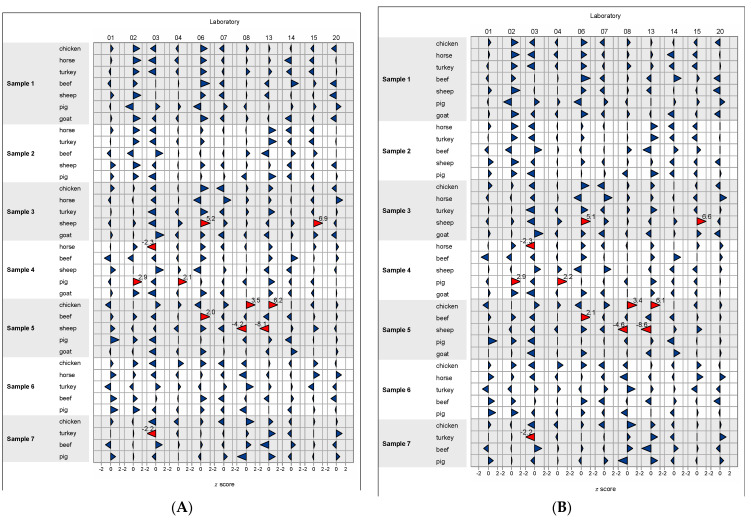
z scores for determination of proportions of animal species. (**A**) AGES database, (**B**) NCBI database. Absolute z scores < 2 are shown in blue, absolute z scores > 2 in red.

**Figure 6 foods-11-01108-f006:**
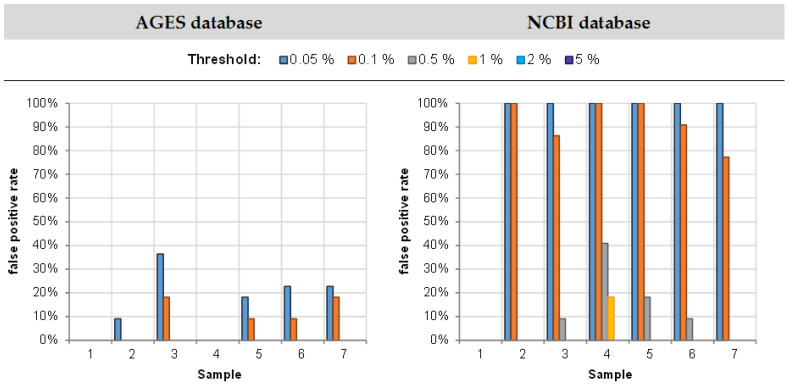
False positive rates by sample, depending on the defined threshold for the AGES (**left**) and the NCBI (**right**) database.

**Figure 7 foods-11-01108-f007:**
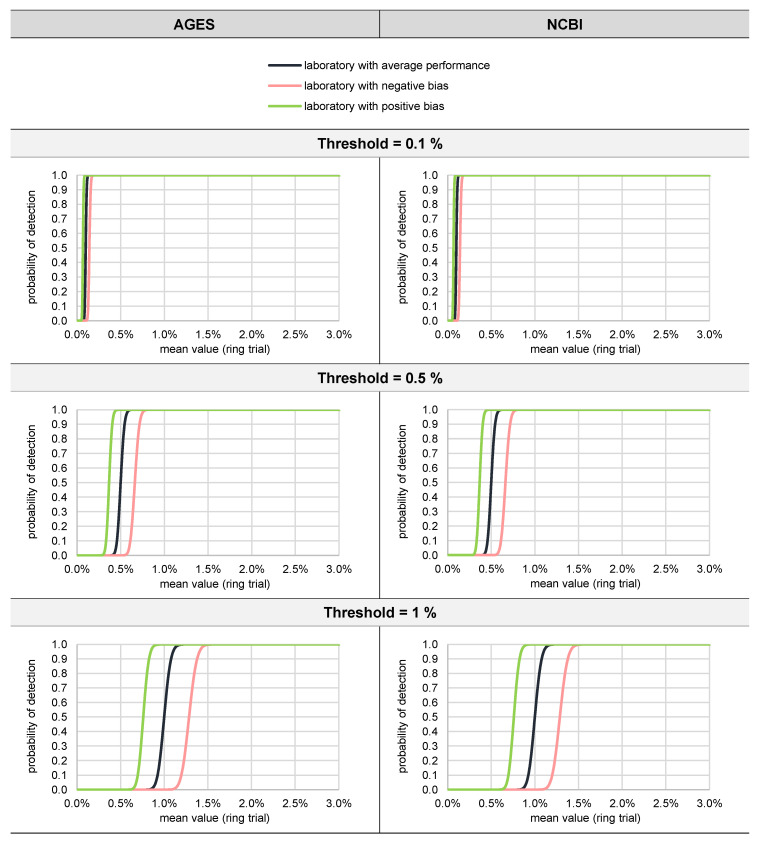
Probability of detection for a laboratory with average performance, a laboratory with positive bias, and a laboratory with negative bias, depending on the threshold and the database.

**Table 1 foods-11-01108-t001:** Sample composition. Samples 1–6: DNA extract mixtures; percentage refers to DNA (*v*/*v*). Sample 7: extract from a model sausage; percentage refers to meat content (*w*/*w*). Sample 8: DNA extract from maize (negative control). - indicates that the species is not in the DNA extract mixture/sample.

Sample	Chicken	Horse	Turkey	Beef	Sheep	Pig	Goat
	Percentage (%)						
1	1	1	1	1	1	94	1
2	-	1.9	0.5	65.7	1.9	30	-
3	1.9	66.1	1.9	-	0.5	-	30
4	-	0.5	-	30	67.5	0.1	1.9
5	67.5	-	-	1.9	30	0.5	0.1
6	0.1	30	67.5	0.5	-	1.9	-
7	5	-	5	50	-	40	-
8	-	-	-	-	-	-	-

**Table 2 foods-11-01108-t002:** Proportion of animal species determined for samples 1 (**a**), 2 (**b**), and 7 (**c**). A and B refer to subsamples A and B, respectively.

**(a). Sample 1**											
**Species**	**Spiking** **Level (%)**	**Laboratory**	**AGES Database**		**NCBI Database**		**Laboratory**	**AGES Database**		**NCBI Database**	
			**A**	**B**	**A**	**B**		**A**	**B**	**A**	**B**
Pig	94	01	91.25	91.77	91.21	91.74	10	95.58	94.83	95.56	94.81
Chicken	1		1.42	1.37	1.42	1.37		0.73	1.08	0.73	1.08
Horse	1		1.42	1.34	1.42	1.33		0.60	0.67	0.60	0.66
Turkey	1		1.17	1.03	1.16	1.02		0.49	0.66	0.41	0.56
Beef	1		1.55	1.51	1.55	1.51		0.57	0.58	0.57	0.58
Sheep	1		1.65	1.57	1.62	1.56		1.05	1.15	1.05	1.15
Goat	1		1.54	1.42	1.56	1.44		0.97	1.03	0.98	1.04
Pig	94	02	90.31	90.33	90.29	90.30	11	94.62	94.90	94.59	94.87
Chicken	1		1.50	1.57	1.50	1.57		0.47	0.30	0.47	0.30
Horse	1		1.64	1.59	1.64	1.59		1.01	1.09	1.00	1.09
Turkey	1		1.32	1.41	1.31	1.41		0.54	0.43	0.51	0.40
Beef	1		1.77	1.67	1.77	1.67		1.02	1.03	1.02	1.03
Sheep	1		1.81	1.73	1.79	1.72		1.19	1.16	1.19	1.16
Goat	1		1.66	1.69	1.68	1.71		1.15	1.10	1.16	1.11
Pig	94	03	92.67	92.48	92.65	92.46	12	94.30	94.68	94.27	94.65
Chicken	1		1.04	1.11	1.04	1.11		0.56	0.36	0.57	0.36
Horse	1		1.04	1.14	1.04	1.14		1.10	1.13	1.10	1.13
Turkey	1		0.90	0.90	0.89	0.89		0.66	0.48	0.64	0.47
Beef	1		1.60	1.55	1.60	1.55		0.98	1.03	0.98	1.04
Sheep	1		1.44	1.48	1.43	1.46		1.24	1.19	1.24	1.19
Goat	1		1.31	1.34	1.32	1.35		1.16	1.13	1.16	1.14
Pig	94	04	92.08	92.28	92.06	92.25	13	92.01	91.80	91.98	91.77
Chicken	1		1.16	1.21	1.16	1.21		1.27	1.33	1.27	1.33
Horse	1		1.22	1.14	1.22	1.14		1.35	1.38	1.35	1.38
Turkey	1		1.09	1.00	1.09	0.99		1.22	1.17	1.22	1.16
Beef	1		1.56	1.56	1.56	1.56		1.45	1.52	1.45	1.52
Sheep	1		1.45	1.49	1.45	1.48		1.38	1.37	1.38	1.37
Goat	1		1.43	1.33	1.44	1.34		1.32	1.43	1.34	1.44
Pig	94	06	90.36	90.41	90.33	90.38	14	90.44	93.48	90.41	93.46
Chicken	1		1.51	1.53	1.51	1.53		1.57	0.97	1.57	0.97
Horse	1		1.52	1.56	1.51	1.56		1.32	0.92	1.32	0.92
Turkey	1		1.36	1.33	1.36	1.32		1.14	0.82	1.13	0.82
Beef	1		1.87	1.84	1.87	1.84		2.17	1.50	2.17	1.50
Sheep	1		1.72	1.67	1.71	1.66		1.81	1.24	1.80	1.24
Goat	1		1.66	1.66	1.68	1.67		1.56	1.06	1.58	1.07
Pig	94	07	92.48	92.19	92.47	92.17	15	92.22	92.23	92.18	92.20
Chicken	1		1.08	1.17	1.08	1.17		1.19	1.18	1.19	1.18
Horse	1		1.21	1.27	1.21	1.27		1.14	1.21	1.14	1.21
Turkey	1		1.03	1.10	1.02	1.10		0.98	0.96	0.98	0.96
Beef	1		1.51	1.50	1.51	1.50		1.60	1.55	1.60	1.55
Sheep	1		1.36	1.38	1.37	1.38		1.48	1.45	1.47	1.44
Goat	1		1.32	1.39	1.32	1.40		1.39	1.41	1.41	1.42
Pig	94	08	91.31	91.28	91.28	91.25	20	92.59	92.63	92.58	92.61
Chicken	1		1.34	1.40	1.34	1.40		1.03	1.09	1.03	1.09
Horse	1		1.40	1.32	1.40	1.32		1.24	1.28	1.24	1.28
Turkey	1		1.28	1.23	1.27	1.23		1.11	1.03	1.11	1.02
Beef	1		1.68	1.62	1.68	1.62		1.44	1.39	1.44	1.39
Sheep	1		1.53	1.59	1.52	1.57		1.31	1.30	1.31	1.29
Goat	1		1.46	1.56	1.48	1.58		1.27	1.28	1.28	1.29
Pig	94	09	94.62	93.61	94.60	93.59					
Chicken	1		0.64	1.07	0.64	1.08					
Horse	1		1.06	1.19	1.06	1.19					
Turkey	1		0.63	1.10	0.61	1.07					
Beef	1		0.92	0.90	0.92	0.90					
Sheep	1		1.10	1.05	1.10	1.05					
Goat	1		1.04	1.07	1.05	1.08					
**(b). Sample 2**											
**Species**	**Spiking** **Level (%)**	**Laboratory**	**AGES Database**		**NCBI Database**		**Laboratory**	**AGES Database**		**NCBI Database**	
			**A**	**B**	**A**	**B**		**A**	**B**	**A**	**B**
Beef	65.7	01	64.59	65.91	64.42	65.73	10	53.26	55.43	53.14	55.32
Pig	30		30.04	29.36	29.93	29.24		41.44	40.09	41.25	39.91
Horse	1.9		2.24	1.97	2.23	1.96		2.21	1.63	2.19	1.61
Sheep	1.9		2.61	2.33	2.58	2.30		2.89	2.71	2.88	2.70
Turkey	0.5		0.46	0.40	0.45	0.39		0.19	0.13	0.16	0.11
Beef	65.7	02	62.99	63.18	62.83	63.01	11	66.01	65.78	65.89	65.68
Pig	30		31.30	31.24	31.20	31.12		29.52	29.75	29.40	29.63
Horse	1.9		2.49	2.44	2.48	2.43		2.03	2.00	2.02	1.98
Sheep	1.9		2.60	2.56	2.57	2.53		2.16	2.27	2.15	2.26
Turkey	0.5		0.61	0.54	0.60	0.53		0.28	0.20	0.26	0.18
Beef	65.7	03	71.67	70.55	71.48	70.36	12	66.29	64.97	66.22	64.88
Pig	30		24.75	25.50	24.68	25.43		29.13	30.25	28.98	30.08
Horse	1.9		1.37	1.44	1.36	1.44		2.08	2.23	2.06	2.21
Sheep	1.9		1.90	2.16	1.88	2.13		2.23	2.31	2.22	2.30
Turkey	0.5		0.30	0.33	0.29	0.33		0.27	0.24	0.25	0.23
Beef	65.7	04	67.22	67.55	67.05	67.39	13	62.21	61.73	62.00	61.52
Pig	30		28.26	28.10	28.17	28.01		32.37	32.75	32.26	32.64
Horse	1.9		1.90	1.76	1.89	1.75		2.52	2.55	2.51	2.54
Sheep	1.9		2.18	2.16	2.17	2.14		2.31	2.31	2.30	2.30
Turkey	0.5		0.40	0.37	0.39	0.37		0.56	0.62	0.55	0.62
Beef	65.7	06	66.46	65.45	66.28	65.24	14	70.20	68.65	69.99	68.47
Pig	30		28.63	29.35	28.54	29.24		26.07	27.18	25.99	27.11
Horse	1.9		2.09	2.10	2.08	2.09		1.44	1.63	1.43	1.62
Sheep	1.9		2.32	2.54	2.31	2.52		1.94	2.15	1.92	2.13
Turkey	0.5		0.47	0.52	0.47	0.52		0.32	0.36	0.32	0.36
Beef	65.7	07	65.64	66.96	65.46	66.77	15	69.77	67.63	69.61	67.48
Pig	30		29.78	28.89	29.68	28.80		26.43	27.98	26.36	27.90
Horse	1.9		1.93	1.73	1.92	1.72		1.49	1.77	1.48	1.76
Sheep	1.9		2.17	2.04	2.17	2.03		1.96	2.20	1.93	2.16
Turkey	0.5		0.47	0.37	0.47	0.37		0.31	0.37	0.31	0.37
Beef	65.7	08	68.90	69.26	68.70	69.06	20	68.01	65.33	67.81	65.14
Pig	30		26.45	26.12	26.36	26.03		28.10	30.10	28.02	30.01
Horse	1.9		1.92	1.85	1.91	1.84		1.69	2.06	1.69	2.05
Sheep	1.9		2.26	2.31	2.23	2.28		1.85	2.09	1.84	2.08
Turkey	0.5		0.43	0.43	0.43	0.43		0.34	0.42	0.34	0.42
Beef	65.7	09	63.64	63.11	63.64	63.11					
Pig	30		31.59	32.21	31.38	31.99					
Horse	1.9		2.26	2.21	2.24	2.18					
Sheep	1.9		2.26	2.25	2.24	2.23					
Turkey	0.5		0.24	0.22	0.23	0.21					
**(c). Sample 7**											
**Species**	**Spiking** **Level (%)**	**Laboratory**	**AGES Database**		**NCBI Database**		**Laboratory**	**AGES Database**		**NCBI Database**	
			**A**	**B**	**A**	**B**		**A**	**B**	**A**	**B**
Beef	50	01	45.79	45.80	45.75	45.78	10	34.33	34.58	34.34	34.59
Pig	40		42.31	42.71	42.25	42.64		61.16	60.60	61.03	60.47
Chicken	5		7.56	7.35	7.55	7.35		3.32	3.46	3.32	3.45
Turkey	5		4.31	4.08	4.28	4.05		1.16	1.33	0.95	1.13
Beef	50	02	48.54	48.18	48.50	48.16	11	45.20	43.03	45.19	43.03
Pig	40		40.68	40.46	40.62	40.40		50.41	45.73	50.30	45.63
Chicken	5		6.71	6.77	6.70	6.76		2.52	6.59	2.52	6.59
Turkey	5		3.99	4.26	3.97	4.25		1.81	4.60	1.70	4.35
Beef	50	03	52.15	52.30	52.08	52.24	12	45.87	42.72	45.91	42.76
Pig	40		38.50	38.68	38.45	38.64		49.65	44.84	49.50	44.72
Chicken	5		6.10	5.87	6.09	5.86		2.58	7.14	2.58	7.13
Turkey	5		3.24	3.13	3.20	3.10		1.84	5.24	1.78	5.08
Beef	50	04	49.31	49.59	49.29	49.57	13	44.13	43.46	44.08	43.41
Pig	40		40.47	40.16	40.42	40.11		43.46	44.09	43.40	44.03
Chicken	5		6.23	6.21	6.22	6.20		7.62	7.44	7.61	7.43
Turkey	5		3.70	3.77	3.69	3.76		4.63	4.85	4.62	4.84
Beef	50	06	50.05	49.52	50.01	49.46	14	50.95	51.46	50.90	51.40
Pig	40		38.40	39.33	38.35	39.27		38.49	38.71	38.44	38.66
Chicken	5		7.42	7.12	7.41	7.11		6.75	6.22	6.74	6.22
Turkey	5		4.08	4.00	4.07	3.99		3.78	3.59	3.76	3.58
Beef	50	07	47.44	48.52	47.39	48.47	15	49.53	48.78	49.49	48.74
Pig	40		42.22	41.83	42.18	41.78		39.38	39.84	39.34	39.80
Chicken	5		6.21	5.89	6.20	5.88		7.03	7.24	7.02	7.24
Turkey	5		4.09	3.72	4.08	3.72		4.02	4.13	4.00	4.11
Beef	50	08	51.82	51.73	51.75	51.67	20	46.99	46.15	46.94	46.11
Pig	40		35.16	36.23	35.11	36.18		41.48	42.03	41.45	42.00
Chicken	5		8.61	7.82	8.59	7.81		6.84	7.18	6.84	7.18
Turkey	5		4.38	4.18	4.37	4.16		4.66	4.61	4.65	4.60
Beef	50	09	44.59	43.04	44.70	43.14					
Pig	40		50.65	51.23	50.45	51.03					
Chicken	5		2.87	3.47	2.86	3.46					
Turkey	5		1.87	2.22	1.80	2.15					

**Table 3 foods-11-01108-t003:** Summary of eliminated outliers.

Sample	Species	Laboratory	Reason
1	All species (*n* = 7)	14	Excessive variance of results for both subsamples
3	Sheep	06	Too high laboratory mean value
		15	Too high laboratory mean value
4	Pig	02	Excessive variance of results for both subsamples
5	Chicken	08	Too high laboratory mean value
		13	Too high laboratory mean value
	Sheep	08	Too low laboratory mean value
		13	Too low laboratory mean value
6	Chicken	15	Excessive variance of results for both subsamples

**Table 4 foods-11-01108-t004:** Statistical parameters for relative proportions of animal species according to ASU §64 LFGB. AGES: AGES database, NCBI: NCBI database.

		Sample													
Species	Parameter ^1^	1		2		3		4		5		6		7	
		AGES	NCBI	AGES	NCBI	AGES	NCBI	AGES	NCBI	AGES	NCBI	AGES	NCBI	AGES	NCBI
Chicken	Number of labs	11	11			11	11			11	11	11	11	11	11
	Number of labs after outlier elimination	10	10			11	11			9	9	10	10	11	11
	Proportion (v/m)	1%				1.9%				67.5%		0.1%		5%	
	Mean value	1.27%	1.26%			2.12%	2.11%			55.49%	55.34%	0.15%	0.15%	6.89%	6.88%
	s_R_	0.18%	0.18%			0.23%	0.23%			0.84%	0.84%	0.03%	0.03%	0.72%	0.71%
	s_r_	0.04%	0.04%			0.08%	0.08%			0.57%	0.57%	0.01%	0.01%	0.24%	0.24%
	Logit proportion	−4.60	−4.60			−3.94	−3.94			0.73	0.73	−6.91	−6.91	−2.94	−2.94
	Logit mean value	−4.36	−4.36			−3.83	−3.83			0.22	0.21	−6.53	−6.53	−2.60	−2.61
	Logit s_r_	0.14	0.14			0.11	0.11			0.03	0.03	0.21	0.21	0.11	0.11
	Logit s_r_	0.03	0.03			0.04	0.04			0.02	0.02	0.05	0.05	0.04	0.04
Horse	Number of labs	11	11	11	11	11	11	11	11			11	11		
	Number of labs after outlier elimination	10	10	11	11	11	11	11	11			11	11		
	Proportion (v/m)	1%		1.9%		66.1%		0.5%				30%			
	Mean value	1.31%	1.31%	1.89%	1.89%	60.3%	60.0%	0.47%	0.47%			42.0%	41.9%		
	s_R_	0.17%	0.17%	0.37%	0.37%	1.1%	1.1%	0.08%	0.08%			1.9%	1.9%		
	s_r_	0.05%	0.05%	0.14%	0.14%	0.3%	0.3%	0.04%	0.04%			0.7%	0.7%		
	Logit proportion	−4.60	−4.60	−3.94	−3.94	0.67	0.67	−5.29	−5.29			−0.85	−0.85		
	Logit mean value	−4.32	−4.32	−3.95	−3.95	0.42	0.40	−5.35	−5.36			−0.32	−0.33		
	Logit s_r_	0.13	0.13	0.20	0.20	0.05	0.05	0.18	0.17			0.08	0.08		
	Logit s_r_	0.04	0.04	0.08	0.08	0.01	0.01	0.08	0.08			0.03	0.03		
Turkey	Number of labs	11	11	11	11	11	11					11	11	11	11
	Number of labs after outlier elimination	10	10	11	11	11	11					11	11	11	11
	Proportion (v/m)	1%		0.5%		1.9%						67.5%		5%	
	Mean value	1.12%	1.12%	0.42%	0.42%	1.86%	1.85%					54.0%	53.8%	4.03%	4.02%
	s_R_	0.16%	0.16%	0.09%	0.09%	0.19%	0.19%					2.1%	2.0%	0.46%	0.46%
	s_r_	0.05%	0.05%	0.04%	0.04%	0.07%	0.07%					0.9%	0.9%	0.14%	0.14%
	Logit proportion	−4.60	−4.60	−5.29	−5.29	−3.94	−3.94					0.73	0.73	−2.94	−2.94
	Logit mean value	−4.48	−4.48	−5.47	−5.48	−3.97	−3.97					0.16	0.15	−3.17	−3.17
	Logit s_r_	0.14	0.14	0.22	0.22	0.10	0.10					0.08	0.08	0.12	0.12
	Logit s_r_	0.05	0.05	0.10	0.10	0.04	0.04					0.04	0.04	0.04	0.04
Beef	Number of labs	11	11	11	11			11	11	11	11	11	11	11	11
	Number of labs after outlier elimination	10	10	11	11			11	11	11	11	11	11	11	11
	Proportion (v/m)	1%		65.7%				30%		1.9%		0.5%		50%	
	Mean value	1.58%	1.58%	66.9%	66.7%			33.02%	32.87%	2.79%	2.79%	0.91%	0.91%	48.7%	48.7%
	s_R_	0.13%	0.13%	2.8%	2.8%			0.55%	0.56%	0.20%	0.20%	0.09%	0.09%	2.6%	2.6%
	s_r_	0.04%	0.04%	1.0%	1.0%			0.31%	0.31%	0.10%	0.10%	0.05%	0.05%	0.4%	0.4%
	Logit proportion	−4.60	−4.60	0.65	0.65			−0.85	−0.85	−3.94	−3.94	−5.29	−5.29	0	0
	Logit mean value	−4.13	−4.13	0.70	0.69			−0.71	−0.71	−3.55	−3.55	−4.69	−4.69	−0.05	−0.05
	Logit s_r_	0.08	0.08	0.13	0.13			0.02	0.03	0.07	0.07	0.10	0.10	0.11	0.10
	Logit s_r_	0.02	0.02	0.04	0.04			0.01	0.01	0.04	0.04	0.06	0.06	0.02	0.02
Sheep	Number of labs	11	11	11	11	11	11	11	11	11	11				
	Number of labs after outlier elimination	10	10	11	11	9	9	11	11	9	9				
	Proportion (v/m)	1%		1.9%		0.5%		67.5%		30%					
	Mean value	1.50%	1.49%	2.21%	2.19%	0.14%	0.14%	64.41%	63.67%	40.92%	40.62%				
	s_R_	0.15%	0.14%	0.22%	0.22%	0.01%	0.01%	0.65%	0.60%	0.62%	0.65%				
	s_r_	0.03%	0.03%	0.14%	0.13%	0.01%	0.01%	0.27%	0.27%	0.49%	0.46%				
	Logit proportion	−4.60	−4.60	−3.94	−3.94	−5.29	−5.29	0.73	0.73	−0.85	−0.85				
	Logit mean value	−4.18	−4.19	−3.79	−3.80	−6.58	−6.60	0.59	0.56	−0.37	−0.38				
	Logit s_r_	0.10	0.10	0.10	0.10	0.10	0.09	0.03	0.03	0.03	0.03				
	Logit s_r_	0.02	0.02	0.06	0.06	0.08	0.08	0.01	0.01	0.02	0.02				
Pig	Number of labs	11	11	11	11			11	11	11	11	11	11	11	11
	Number of labs after outlier elimination	10	10	11	11			10	10	11	11	11	11	11	11
	Proportion (v/m)	94%		30%				0.1%		0.5%		1.9%		40%	
	Mean value	91.77%	91.74%	28.5%	28.4%			0.04%	0.04%	0.61%	0.61%	2.90%	2.90%	40.2%	40.1%
	s_R_	0.83%	0.83%	2.2%	2.2%			0.02%	0.02%	0.12%	0.12%	0.26%	0.26%	2.3%	2.3%
	s_r_	0.16%	0.16%	0.7%	0.7%			0%	0%	0.04%	0.04%	0.13%	0.13%	0.4%	0.4%
	Logit proportion	2.75	2.75	−0.85	−0.85			−6.91	−6.91	−5.29	−5.29	−3.94	−3.94	−0.41	−0.41
	Logit mean value	2.41	2.41	−0.92	−0.92			−7.73	−7.74	−5.09	−5.10	−3.51	−3.51	−0.40	−0.40
	Logit s_r_	0.11	0.11	0.11	0.11			0.41	0.41	0.19	0.19	0.09	0.09	0.10	0.10
	Logit s_r_	0.02	0.02	0.03	0.03			0.10	0.10	0.07	0.07	0.05	0.05	0.02	0.02
Goat	Number of labs	11	11			11	11	11	11	11	11				
	Number of labs after outlier elimination	10	10			11	11	11	11	11	11				
	Proportion (v/m)	1%				30%		1.9%		0.1%					
	Mean value	1.44%	1.45%			35.4%	35.6%	2.00%	2.01%	0.09%	0.09%				
	s_R_	0.14%	0.14%			1.0%	1.0%	022%	0.23%	0.02%	0.02%				
	s_r_	0.05%	0.05%			0.3%	0.3%	0.10%	0.10%	0.02%	0.02%				
	Logit proportion	−4.60	−4.60			0.85	−0.85	−3.94	−3.94	−6.91	−6.91				
	Logit mean value	−4.23	−4.22			−0.60	−0.59	−3.89	−3.89	−7.05	−6.99				
	Logit s_r_	0.10	0.10			0.04	0.05	0.11	0.11	0.19	0.19				
	Logit s_r_	0.04	0.04			0.01	0.01	0.05	0.05	0.19	0.17				

1 s_R_: reproducibility standard deviation; s_r_: repeatability standard deviation; v: volume; m: mass.

**Table 5 foods-11-01108-t005:** Bias standard deviation, reproducibility standard deviation, and repeatability standard deviation (absolute, i.e., retransformed to proportions of animal species) depending on the proportion of animal species.

Standard Deviation	Proportion of Animal Species	
	5%/95%	50%
Absolute bias standard deviation	1.8%	7.2%
Absolute reproducibility standard deviation	0.5%	1.8%
Absolute repeatability standard deviation	0.2%	0.6%

**Table 6 foods-11-01108-t006:** False positive reads obtained for samples 01–07. (**A**): AGES database, (**B**): NCBI database. Laboratories 01–06, 08, 14, 15: MiSeq; laboratories 07, 13, 20: iSeq 100.

**(A)**												
**Sample**	**Subsample**	**Laboratory**										
		**01**	**02**	**03**	**04**	**06**	**07**	**08**	**13**	**14**	**15**	**20**
1	A	6	2	1	-	2	-	2	-	2	2	-
	B	2	5	-	3	3	-		-	1	4	-
2	A	99	41	21	95	78	6	76	41	44	87	5
	B	43	89	31	121	73	8	74	43	34	121	
3	A	49	325	14	1234	177	91	47	359	91	88	64
	B	51	369	40	1356	196	85	44	355	79	52	35
4	A	55	5	12	38	106	69	57	3	90	57	10
	B	38	34	20	64	117	37	80	10	100	67	10
5	A	6	51	2	286	173	121	56	124	271	79	65
	B	9	57	3	286	212	62	66	137	396	39	32
6	A	8	37	2	111	274	80	110	33	161	41	64
	B	5	50	7	102	314	46	106	50	169	71	27
7	A	45	188	42	608	99	62	69	250	61	62	39
	B	80	717	42	596	140	83	86	294	49	28	25
**(B)**												
**Sample**	**Subsample**	**Laboratory**										
		**01**	**02**	**03**	**04**	**06**	**07**	**08**	**13**	**14**	**15**	**20**
1	A	104	75	122	49	72	40	65	20	48	72	19
	B	120	91	164	59	81	24	58	35	29	54	25
2	A	630	715	612	743	804	242	891	465	526	637	388
	B	489	783	1047	753	898	388	752	540	432	682	345
3	A	387	766	456	1639	497	394	209	583	355	297	242
	B	241	798	633	1792	606	383	229	621	365	177	127
4	A	2133	400	2738	1482	1908	1095	2214	670	1164	2428	585
	B	1584	2043	2896	1717	1941	1477	2182	837	1390	2585	668
5	A	987	846	1963	1127	1259	1177	988	531	863	924	487
	B	1711	1179	2261	1063	1368	468	1202	628	1165	547	297
6	A	501	590	2354	634	796	584	556	437	525	371	261
	B	627	580	2619	584	853	394	518	440	612	543	200
7	A	250	452	496	788	322	210	426	459	244	229	175
	B	260	950	422	819	627	366	388	519	245	192	132

**Table 7 foods-11-01108-t007:** Proportion of results below a thresholds of 0.05%, 0.1%, 0.5%, and 1%.

Sample	Species	Spiking Level (%)	Proportion of Results below a Threshold of							
			0.05%	0.1%	0.5%	1%	0.05%	0.1%	0.5%	1%
			AGES Database				NCBI Database			
1	Pig	94	-	-	-	-	-	-	-	-
	Chicken	1	-	-	-	1/22 (5%)	-	-	-	1/22 (5%)
	Horse	1	-	-	-	1/22 (5%)	-	-	-	1/22 (5%)
	Turkey	1	-	-	-	5/22 (23%)	-	-	-	6/22 (27%)
	Beef	1	-	-	-	-	-	-	-	-
	Sheep	1	-	-	-	-	-	-	-	-
	Goat	1	-	-	-	-	-	-	-	-
2	Beef	65.7	-	-	-	-	-	-	-	-
	Pig	30.0	-	-	-	-	-	-	-	-
	Horse	1.9	-	-	-	-	-	-	-	-
	Sheep	1.9	-	-	-	-	-	-	-	-
	Turkey	0.5	-	-	17/22 (77%)	22/22 (100%)	-	-	17/22 (77%)	22/22 (100%)
3	Horse	66.1	-	-	-	-	-	-	-	-
	Goat	30.0	-	-	-	-	-	-	-	-
	Chicken	1.9	-	-	-	-	-	-	-	-
	Turkey	1.9	-	-	-	-	-	-	-	-
	Sheep	0.5	-	-	22%22 (100%)	22/22 (100%)	-	-	22/22 (100%)	22/22 (100%)
4	Sheep	67.5	-	-	-	-	-	-	-	-
	Beef	30.0	-	-	-	-	-	-	-	-
	Goat	1.9	-	-	-	-	-	-	-	-
	Horse	0.5	-	-	13/22 (59%)	22/22 (100%)	-	-	14/22 (64%)	22/22 (100%)
	Pig	0.1	14/22 (64%)	20/22 (91%)	22/22 (100%)	22/22 (100%)	14/22 (64%)	20/22 (91%)	22/22 (100%)	22/22 (100%)
5	Chicken	67.5	-	-	-	-	-	-	-	-
	Sheep	30.0	-	-	-	-	-	-	-	-
	Beef	1.9	-	-	-	-	-	-	-	-
	Pig	0.5	-	-	4/22 (18%)	22/22 (100%)	-	-	4/22 (18%)	22/22 (100%)
	Goat	0.1	-	15/22 (68%)	22/22 (100%)	22/22 (100%)	-	12/22 (55%)	22/22 (100%)	22/22 (100%)
6	Turkey	67.5	-	-	-	-	-	-	-	-
	Horse	30.0	-	-	-	-	-	-	-	-
	Pig	1.9	-	-	-	-	-	-	-	-
	Beef	0.5	-	-	-	19/22 (86%)	-	-	-	19/22 (86%)
	Chicken	0.1	-	-	22/22 (100%)	22/22 (100%)	-	-	22/22 (100%)	22/22 (100%)
7	Beef	50.0	-	-	-	-	-	-	-	-
	Pig	40.0	-	-	-	-	-	-	-	-
	Chicken	5.0	-	-	-	-	-	-	-	-
	Turkey	5.0	-	-	-	-	-	-	-	-

**Table 8 foods-11-01108-t008:** Reads obtained for sample 8 (negative control). (**A**): AGES database, (**B**): NCBI database.

**(A)**																				
**Species**	**Laboratory**																			
	**01**		**02**		**03**		**04**		**06**		**07**		**08**		**14**		**15**		**20**	
	**A**	**B**	**A**	**B**	**A**	**B**	**A**	**B**	**A**	**B**	**A**	**B**	**A**	**B**	**A**	**B**	**A**	**B**	**A**	**B**
Chicken	-	-	13	28	-	-	42	30	4	98	-	-	2	-	10	177	2	10	8	5
Horse	-	-	7	6	-	2	57	51	-	2	-	-	-	-	-	1	-	-	7	5
Turkey	-	-	5	11	-	6	51	52	3	7	1	-	-	3	232	6	6	15	13	9
Beef	10	9	25	22	2	171	17	65	101	97	222	-	43	51	185	148	57	99	110	55
Pig	-	1	1	3	23	87	11	3	216	114	563	-	21	21	91	56	38	54	50	26
Sheep	5	14	51	79	2	2	82	74	3	9	19	-	3	1	11	3	-	-	9	10
Goat	1	1	-	1	-	-	1	1	1	1	2	-	-	-	-	1	-	-	-	1
*Bison bonasus*	-	-	-	-	-	-	-	-	-	-	-	-	-	1	-	2	-	-	-	-
Total reads	16	25	102	150	27	268	261	276	328	328	807	-	69	77	529	394	103	178	197	111
**(B)**																				
**Species**	**Laboratory**																			
	**01**		**02**		**03**		**04**		**06**		**07**		**08**		**14**		**15**		**20**	
	**A**	**B**	**A**	**B**	**A**	**B**	**A**	**B**	**A**	**B**	**A**	**B**	**A**	**B**	**A**	**B**	**A**	**B**	**A**	**B**
Chicken	-	-	13	28	-	-	42	30	4	98	-	-	2	-	10	177	2	10	8	5
Horse	-	-	7	6	-	2	57	51	-	2	-	-	-	-	-	1	-	-	7	5
Turkey	-	-	5	11	-	6	51	50	3	7	1	-	-	3	231	6	6	15	13	9
Beef	10	9	25	22	2	171	17	65	101	97	222	-	43	52	187	149	57	99	111	55
Pig		1	1	3	23	99	11	3	226	114	564	-	21	21	91	57	38	54	50	27
Sheep	6	14	50	76	2	2	80	74	3	9	19	-	3	1	7	3	-	-	9	10
Goat	1	1	-	1	-	-	1	1	1	1	2	-	-	-	-	1	-	-	-	1
*Bison bonasus*	-	-	-	-	-	-	-	-	-	-	-	-	-	-	-	1	-	-	-	-
*Bos mutus*	-	-	-	-	-	-	-	-	-	1	-	-	-	1	-	-	1	-	-	-
*Brachypodium sylvaticum*	-	-	-	-	-	-	-	-	1	-	-	-	-	-	-	-	-	-	-	-
*Coregonus migratorius*	-	-	-	-	-	-	-	-	2	-	-	-	-	-	-	-	-	-	-	-
*Equus zebra*	-	-	-	-	-	-	-	-	-	-	-	-	-	-	-	-	-	-	-	-
*Eukaryotic synthetic*	-	-	-	15	-	-	-	-	-	-	-	-	-	-	-	-	-	-	-	-
*Homo sapiens*	-	-	58	46	30	188	82	215	302	784	1086	-	5	124	687	451	-	15	-	49
*Meleagris ocellata*	-	-	-	-	-	-	-	2	-	-	-	-	-	-	1	-	-	-	-	-
*Oncorhynchus environmental*	-	-	-	-	-	-	-	-	1	1	-	-	-	-	-	-	-	-	-	-
*Oncorhynchus mykiss*	-	-	-	-	-	-	-	-	37	91	-	-	-	-	-	-	-	-	-	-
*Ovis ammon*	-	1	1	5	-	-	2	1	1	-	-	-	-	-	4	-	-	-	-	-
*Ovis vignei*	-	-	-	-	-	-	1	1	-	-	-	-	-	-	-	-	-	-	-	-
*Phascolosoma esculenta*	-	-	-	-	-	1	-	-	-	-	-	-	-	-	-	-	-	-	-	-
*Synthetic construct*	-	-	-	-	-	-	-	-	20	-	-	-	-	-	-	-	-	-	-	-
*Zea mays*	-	-	-	2	-	-	-	3	851	3	-	-	-	-	-	-	-	-	2	-
total reads	17	26	160	215	57	469	344	496	1553	1208	1894	-	74	202	1218	846	104	193	200	161

## Data Availability

The datasets generated during the current study are available from the corresponding authors upon reasonable request.

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
