# Peer review of "Interlaboratory Validation of a DNA Metabarcoding Assay for Mammalian and Poultry Species to Detect Food Adulteration"

_foods, 2022, doi:10.3390/foods11081108_

Round 1

Reviewer 1 Report

The manuscript focuses on the DNA metabarcoding assay for food authentication, which has been tested in an interlaboratory ring trial including 15 laboratories. The manuscript is well written and organized. However, there are some places which have to be carefully revised by the authors.

Major comments:

  1. Too many tables and figures appear in the manuscript, and some tables and figures could be placed in the Supplementary Data File.
  2. Lin 104-107. Please provide information on the source of the meat.
  3. Line 17-19. In view of the principle of DNA metabarcoding and the required hardware, I don't think that the DNA metabarcoding can replace technologies such as real-time PCR in the near future.
  4. Line 108-112. Why was pure DNA mixing of different species chosen as the models rather than direct mixing of meat? Models of meat mixed in different qualities seem to be closer to reality.
  5. Line 346. What are the criteria for defining outliers? Please give a detailed explanation.
  6. Line 591. Please give clear conclusions rather than results and discussions. Please rewrite the conclusion.

Minor corrections:

Line 105, 109. correct beef to cattle.

Line 117, 479, 500, 519, 577, 581. please note the correct format of the tables.

Line 194. “Supplementary table S1” Please provide supplementary data files.

Line 288, 297. “p < 0001” Please confirm it is correct.

Reviewer 2 Report

Summary

The present study describes the results from an interlaboratorial assay of DNA metabarcoding method for the identification of 15 mammalian and six poultry species simultaneously. The trial described aimed at the harmonization of analytical methods for food authentication across EU Member States. It included 15 laboratories who analyzed 16 anonymously labelled samples. Data was statistically processed for the assessment of repeatability, reproducibility, robustness and uncertainty. This work is original, involves a lot of work and adequate data analysis and is certainly a good contribution to widen the use of NGS in food authentication.

General concept comments

This publication is relevant for food quality control and food authenticity laboratories since the metabarcoding is a methodology in permanent expansion due to price decay and will enable high-throughput assays. This methodology presents a great advantage over classical barcodes sequencing as it enables a certain degree of untargeted analysis which can identify species not expected to be used as adulterants of certain products.

Specific comments

Line 43: Authors stated that: “In the past, authentication of meat products in official food laboratories was mainly based on real-time polymerase chain reaction (PCR) assays and/or DNA arrays.” In my opinion this sentence should be rephrased as qPCR, DNA arrays and barcode sequencing are still the most used the techniques currently in official labs.

Line 140: Concerns related to the relation of copy number with the mass of meat should be addressed. The method is useful for the relative % of each specie determination. However, as other DNA based methods, the accurate absolute quantification of the mass of meat used to prepare the mixtures is not possible since the copy number of target genes in a sample of meat depends on several factors as, for instance, the water content of sample. This is not to be consider a limitation of the method however should be referred in the manuscript.

Paragraph from lines 167-172 would be better fitted before line 153.

Round 2

Reviewer 1 Report

The authors have made all the corrections to the manuscript. The paper is now in conditions to be published in the journal.